# Volume-to-area law entanglement transition in a non-Hermitian free fermionic chain

**Youenn Le Gal, Xhek Turkeshi and Marco Schirò⋆**

JEIP, USR 3573 CNRS, Collège de France, PSL Research University,
11 Place Marcelin Berthelot, 75321 Paris Cedex 05, France

⋆ marco.schiro@college-de-france.fr

## Abstract

We consider the dynamics of the non-Hermitian Su-Schrieffer-Heeger model arising as the no-click limit of a continuously monitored free fermion chain where particles and holes are measured on two sublattices. The model has $\mathcal{PT}$-symmetry, which we show to spontaneously break as a function of the strength of measurement backaction, resulting in a spectral transition where quasiparticles acquire a finite lifetime in patches of the Brillouin zone. We compute the entanglement entropy's dynamics in the thermodynamic limit and demonstrate an entanglement transition between volume-law and area-law scaling, which we characterize analytically. Interestingly we show that the entanglement transition and the $\mathcal{PT}$-symmetry breaking do not coincide, the former occurring when the entire decay spectrum of the quasiparticle becomes gapped.

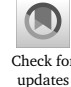

# 1 Introduction

Unitarity reflects core aspects of quantum mechanics, such as probability conservation and the linearity of the Schrödinger equation [1], and constrains how quantum information and correlations spread in a many-body system, as demonstrated by the extensive study of non-equilibrium dynamics of isolated quantum many-body systems [2–6].

However in several instances of theoretical and experimental relevance one is interested in non-unitary quantum dynamics. A notable example is the recent interest on monitored quantum many-body systems, where unitary evolution competes with quantum measurements, resulting in a rich pattern of entanglement and correlation dynamics. In this context several works have highlighted a measurement-induced entanglement phase transition, separating an error-correcting phase, in which the quantum information encoded in the system is resilient to the external probes, and a quantum Zeno phase, where frequent measurements continuously collapse the dynamics to a restricted manifold [7–36]. The scaling of entanglement at the stationary state characterizes the dynamical phases and their transitions. For example, random quantum circuits follow a volume-law scaling in the error-correcting phase, while it has an area-law behavior in the quantum Zeno phase [37–39]. Conversely, monitored free fermionic systems typically follow a logarithmic entanglement scaling in the error-correcting phase [40–49]. This phenomenon is considered an effect of their Gaussian nature, leading to more drastic measurement effects on the system [50].

Another example of non-unitarity is provided by non-Hermitian quantum mechanics, which has a long tradition dating back to works on noninteracting electronic disordered systems [51, 52], to studies of non-unitary conformal field theory [53,54], of few body-systems with parity-time (PT) reversal symmetry [55] or exceptional points [56]. In quantum optics non-Hermitian evolution naturally arises when a measurement apparatus monitors the system, which stems from the measurements and their backaction, and the subsequent post-selection of quantum trajectories corresponding to no-measurement events (no-click limit) [57–61].

Recently non-Hermitian quantum many-body systems are attracting widespread interest for their exotic properties, including quantum criticality and topology (see Refs. [62,63] for a comprehensive review). Their dynamical behavior has also received attention recently, in particular for $\mathcal{PT}$-symmetric systems such as Luttinger Liquids [64,65] or non-interacting lattice models tuned at an exceptional point. In this latter case the entanglement entropy has been shown to grow linearly in time and to saturate to a volume-law [66–68], similarly to their Hermitian counterpart. In presence of interactions it has been proposed that an entanglement transition from an extensive to a sub-extensive scaling could take place in correspondence with $\mathcal{PT}$-symmetry breaking [69]. Other mechanisms for entanglement transitions have been discussed, such as the non-Hermitian skin effect [70] or a spectral transition between a gapless and a gapped phase in the non-Hermitian XY chain, leading to an entanglement transition between a logarithmic and area-law scaling as the strength of the measurement backaction is tuned [71]. Overall, a full understanding of the entanglement patterns of non-Hermitian systems is still missing and exact calculations on an analytically treatable quantum system can provide important insights.

In this work we reconsider the role of $\mathcal{PT}$-symmetry and its breaking for entanglement transitions in a non-interacting free fermionic systems. Taking a non-Hermitian Su-Schrieffer-Heeger (SSH) model [63,66,67,72,73] as working example we compute exactly its spectral properties and entanglement dynamics in the thermodynamic limit. The resulting dynamical phase diagram, plotted in Fig. 1, shows the existence of a volume-to-area-law entanglement transition as a function of the strength of the measurement backaction. Quite interestingly this turns to be distinct in general from the $\mathcal{PT}$-symmetry breaking transition.

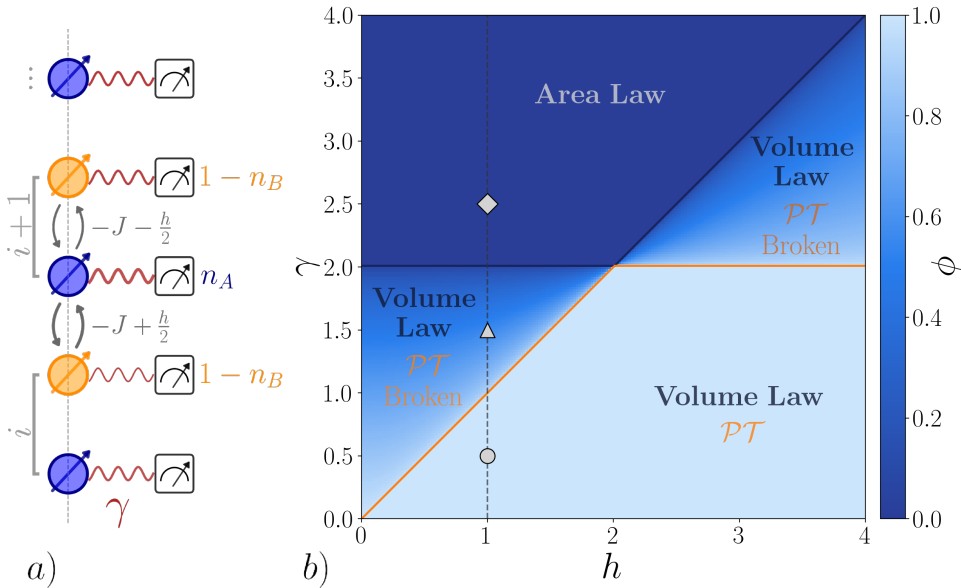

Figure 1: a) Cartoon of the setting under study, giving rise to a non-Hermitian SSH model. A one dimensional chain with two sublattices (A and B, respectively blue and orange circles) and staggered hopping $-J \pm \frac{h}{2}$ is coupled to a monitoring environment measuring particle and holes in the two sublattices. The measurement backaction induces a staggered imaginary chemical potential $\pm i\gamma$ leading to the non-Hermitian SSH in Eq. (1) (See Appendix A). b) Phase diagram in the $(\gamma, h)$ plane summarizing spectral and entanglement transitions in the system. The light blue area corresponds to a purely real spectrum, as indicated by the percentage $\phi$ of real eigenvalues in the spectrum, which is the $\mathcal{PT}$-symmetric phase, and the dark blue one to a purely imaginary spectrum. The gradient area presents a mixed spectrum with both purely real and purely imaginary eigenvalues. Two critical lines emerge, $\gamma_{PT}$ (orange line) and $\gamma_c$ (black line), corresponding respectively to the $\mathcal{PT}$-symmetry breaking and to the opening of a gap in the imaginary part of the spectrum. The entanglement transition from volume to area law scaling occurs in correspondence with the second transition at $\gamma_c$. The three markers will be used in the following figures to characterize the properties of the system.

The remaining is structured as follows. In Sec. 2 we introduce the model and the dynamical protocol we consider, particularly discussing how the non-Hermitian dynamics can arise from full stochastic evolution. In Sec. 3, we diagonalize the model with periodic boundary conditions and discuss its spectrum, showing that it displays a transition associated with a breaking of $\mathcal{PT}$ symmetry. In Sec. 4 we present our results for the entanglement dynamics and the associated entanglement transition from volume to area law. We present a discussion of our results in Sec. 5 and our conclusions and future directions in Sec. 6.

## 2 Non-Hermitian Su-Schrieffer-Heeger Model

This section introduces and discusses the model of interest for this paper: the non-Hermitian SSH chain [63, 66, 67, 72, 73]. We consider a periodic chain made of $L$ units, each one containing two inequivalent sites $A, B$. We assume periodic boundary conditions, and label the fermionic degrees of freedom of the chain with two indices $A$ and $B$ corresponding to the

two sublattices (Fig. 1). We are interested in the quantum dynamics generated by the non-Hermitian Hamiltonian

$$H_{\text{eff}} = H - i\gamma \sum_{i=1}^{L} \left( c_{A,i}^{\dagger} c_{A,i} - c_{B,i}^{\dagger} c_{B,i} \right) . \tag{1}$$

where $H$ describes a conventional SSH chain

$$H = \sum_{j=1}^{L} \left[ -\left( J - \frac{h}{2} \right) c_{A,j}^{\dagger} c_{B,j-1} - \left( J + \frac{h}{2} \right) c_{A,j}^{\dagger} c_{B,j} + \text{h.c.} \right] , \tag{2}$$

with periodic boundary conditions and a staggered hopping for $h \neq 0$, while the last term in Eq. (1) is a purely imaginary staggered chemical potential which accounts for the backaction of a measurement protocol trying to monitor particle and holes in the two sublattices (See Appendix A for a derivation of $H_{\text{eff}}$ from quantum jumps dynamics). The system evolution under $H_{\text{eff}}$ reads as [1]

$$|\Psi(t)\rangle = \frac{e^{-iH_{\text{eff}}t}|\Psi(0)\rangle}{\|e^{-iH_{\text{eff}}t}|\Psi(0)\rangle\|} , \tag{3}$$

where the normalization of the wave function, directly inherited from the quantum jumps protocol as presented in Appendix A, is due to the fact that for generic non-Hermitian systems the norm is not conserved during the evolution because of the lack of unitarity. We note that this is the case also in presence of $\mathcal{PT}$-symmetry due to the non-orthogonality of eigenstates of the non-Hermitian Hamiltonian. The normalization can be implemented by solving the following deterministic evolution

$$d|\Psi(t)\rangle = -iH_{\text{eff}}dt|\Psi(t)\rangle - i\frac{dt}{2}\langle H_{\text{eff}} - H_{\text{eff}}^{\dagger}\rangle_t|\Psi(t)\rangle , \tag{4}$$

corresponding to the no-click limit of the stochastic Schrödinger equation derived in Appendix A.

The Eq. (3) is the central object we will investigate in this manuscript. Throughout this paper, we fix the energy scale $J = 1$ without loss of generality. For convenience, we fix the initial state $|\Psi(0)\rangle$ to be the ground state of $H$ (cf. Eq. (2)) with $h = 0$ (the XX spin chain). Nevertheless, the discussion presented in this paper is extendable to other initial states, provided they have sub-extensive entanglement content.

We conclude by emphasizing that, as we are going to discuss in the next section, the choice of the non-Hermitian SSH model is motivated by the presence of a $\mathcal{PT}$-symmetry which spontaneously breaks as a function of system parameters leading to a real-to-complex eigenvalue transition. This allows us to explore, in a quadratic and thus exactly solvable model, the connections between spectral and entanglement transitions and the role of $\mathcal{PT}$-symmetry breaking. We emphasize that the topological nature of the non-Hermitian SSH model is not particularly relevant for the present work. Other models with $\mathcal{PT}$-symmetry could be considered, such as for example the non-Hermitian Kitaev chain with gain and losses [74].

## 3   Spectral transition and $\mathcal{PT}$-Symmetry Breaking

In this section, we study the spectral properties of the non-Hermitian Hamiltonian in Eq. (1).

A fundamental property of systems described by a non-Hermitian Hamiltonian is that their spectrum is generally complex, with the imaginary part corresponding to excitations with a

---

[1]We study the non unitary dynamics arising from the non-Hermitian effective hamiltonian, it concerns only right eigenvectors, bi-orthogonal quantum mechanics properties will not be used (such as inner product redefinition). We only study how $\mathcal{PT}$-symmetry breaking modifies the non-unitary dynamics.

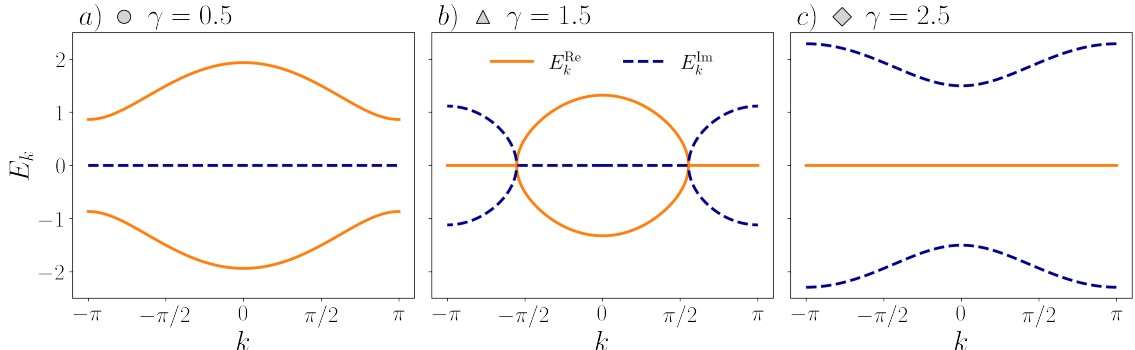

Figure 2: Evolution of the spectrum of the Non-Hermitian SSH model along the line $h = 1$ for three values of $\gamma = 0.5, 1, .5, 2.5$ ( symbols in the top-left corners corresponding to points in the phase diagram of Fig. 1 ). For $\gamma = 0.5$ (panel a) the spectrum is purely real and gapped at $k = \pm\pi$. Above $\gamma_{PT} = h = 1$ the $\mathcal{PT}$-symmetry breaks and a certain fraction of eigenvalues become purely imaginary, corresponding to $k-$modes near the zone boundaries acquiring a finite lifetime. The system has two exceptional points at $k = \pm k_{EP}(\gamma)$. As $\gamma$ increases further above $\gamma_c = 2$ (panel c) the full spectrum becomes purely imaginary and gapped.

finite lifetime. This fact is not surprising since we emphasize that our system is open and in contact with an environment even within the no-click limit. A relevant exception to this paradigm is provided by $\mathcal{PT}$-symmetric non-Hermitian Hamiltonians [55], where a combination of parity ($\mathcal{P}$), i.e. space-reflection, and time-reversal ($\mathcal{T}$) symmetry provides a sufficient condition for the spectrum to be purely real. In the context of our non-Hermitian SSH the action of parity and time-reversal reads respectively

$$\mathcal{P}c_{A,j}\mathcal{P} = c_{B,L-j+1}\,, \qquad \mathcal{P}c_{B,j}\mathcal{P} = c_{A,L-j+1}\,, \qquad \mathcal{T}i\mathcal{T} = -i\,. \tag{5}$$

With these actions $\mathcal{P}^2 = 1$ and $\mathcal{P}$ and $\mathcal{T}$ commutes. The $\mathcal{PT}$ transformation applied to the non-Hermitian SSH Hamiltonian in Eq. (3) then fulfills $(\mathcal{PT})H_{eff}(\mathcal{TP}) = H_{eff}$ which means that $H_{eff}$ presents a $\mathcal{PT}$-symmetry and can have purely real eigenvalues.

The spectrum is purely real when the eigenvectors of $H_{\text{eff}}$ are eigenvectors of the $\mathcal{PT}$-symmetry operation, then the $\mathcal{PT}$-symmetry is unbroken [55] and the system is said to be in a $\mathcal{PT}$-symmetric phase. Conversely, when this condition does not hold, the spectrum presents complex conjugate eigenvalues (which in our case are purely imaginary Fig. 2), and the system is in a $\mathcal{PT}$-symmetry broken phase.

Our non-Hermitian SSH Hamiltonian in Eq. (1) is quadratic which allows us to compute exactly its spectrum. We perform the Fourier transform $c_{X,j} = \sum_k e^{ikj}c_{X,k}/\sqrt{L}$ where the sum of $k$ runs over the Brillouin zone $[-\pi, \pi]$ and $X = A, B$. Neglecting irrelevant constant contributions, we have

$$
\begin{aligned}
H_{\text{eff}} &= \sum_k \begin{pmatrix} c_{A,k}^\dagger & c_{B,k}^\dagger \end{pmatrix} \mathcal{H}_k \begin{pmatrix} c_{A,k} \\ c_{B,k} \end{pmatrix}, \\
\mathcal{H}_k &= \begin{pmatrix} i\gamma & \left(-1 + \frac{h}{2}\right)e^{-ik} - \left(1 + \frac{h}{2}\right) \\ \left(-1 + \frac{h}{2}\right)e^{+ik} - \left(1 + \frac{h}{2}\right) & -i\gamma \end{pmatrix}.
\end{aligned} \tag{6}
$$

The eigenvalues of the single-particle Hamiltonian $\mathcal{H}_k$ fixes the spectral properties of the system, Diagonalizing $\mathcal{H}_k$ we get

$$E_k = \pm\sqrt{h^2 - \gamma^2 + (4 - h^2)\cos^2\left(\frac{k}{2}\right)}\,. \tag{7}$$

As clear in Eq. (7), depending on the values of $\gamma$ and $h$, $E_k$ can be either real or imaginary and the system may be in a $\mathcal{PT}$-symmetric or broken phase. We illustrate the phase diagram in Fig. 1 employing the density of real eigenvalues, defined as

$$\phi = \int_{-\pi}^{\pi} \frac{dk}{2\pi} \Theta\left( h^2 - \gamma^2 + (4 - h^2)\cos^2\left(\frac{k}{2}\right)\right), \tag{8}$$

with $\Theta(x) = 1$ for $x > 0$ and $\Theta(x) = 0$ for $x < 0$ the Heaviside function.

We can fix $h = 1$ and discuss the qualitative features of the spectrum varying $\gamma$ for reference, and discuss how the real and imaginary part of $E_k = E_k^{\text{Re}} + iE_k^{\text{Im}}$ are affected. As we show in Fig. 2 (panel a) for small values of $\gamma$ the spectrum is purely real and has a gap at $k = \pm\pi$ as in the conventional SSH Hermitian case. This regime corresponds to the $\mathcal{PT}$-symmetric phase. As one increases $\gamma$, the gap at the edge of the Brillouin zone decreases and ultimately closes at $\gamma_{PT} = h = 1$, where two exceptional points (EPs) emerge at $k = \pm\pi$ [67, 72]. The $\mathcal{PT}$-symmetry breaking occurs when $\gamma$ is increased further above $\gamma_{PT}$ (panel b) and part of the $k$-modes acquire a finite lifetime given by the imaginary part of the energy $E_k^{\text{Im}}$. In this regime the spectrum is gapless both in its real and imaginary part and has two EPs at $k = \pm k_{EP}(\gamma)$ whose value can be read from the spectrum in Eq. (3) and it is given (for $h < 2$) by

$$k_{EP}(\gamma) = \pm 2\arccos\sqrt{\frac{\gamma^2 - h^2}{4 - h^2}}. \tag{9}$$

As the strength of $\gamma$ is increased further the EPs in Eq. (9) move towards $k = 0$ and finally merge for $\gamma_c = 2$. This signals a second spectral transition above which all the spectrum becomes purely imaginary and gapped (panel c). To summarize, the analysis of the spectrum reveals two separate transitions, where first $\mathcal{PT}$-symmetry breaks and some eigenvalues acquire an imaginary part, and then the full spectrum becomes imaginary and gapped. A qualitatively similar behavior occurs for other values of $h$ and leads to the spectral phase diagram reported in Figure 1. We note that for $h > 2$ the $\mathcal{PT}$-symmetry breaking transition occurs at $\gamma_{PT} = 2$, independently of $h$, while the gapless-to-gapped transition occurs at $\gamma_c = h$ (See Fig. 1). Remarkably, as we are going to show in the next section, the spectral properties of the system reflect the non-trivial entanglement dynamics and stationary states of our model. In particular, we will see how the entanglement transition occurs in correspondence of the critical point $\gamma_c$ where the spectrum becomes fully imaginary and gapped.

## 4 Entanglement Transition

This section discusses the dynamics and stationary-state behavior of the entanglement entropy of the non-Hermitian SSH as a function of system parameters and in particular the emergence of an entanglement transition from volume to area law scaling.

To this extent we first discuss how entanglement entropy can be efficiently computed in free fermionic systems through the correlation matrix. Afterward, we present the analytic computation of the entanglement dynamics and the stationary state entanglement entropy. In particular, we obtain an explicit expression for the leading order entanglement scaling with system size.

## 4.1 Dynamics of the correlation matrix and entanglement entropy

Our non-Hermitian Hamiltonian is quadratic therefore starting from a Gaussian fermionic state, by Gaussianity, the dynamics is entirely encoded in the correlation matrix

$$G_{m,n} = \begin{pmatrix} \langle c^\dagger_{A,m} c_{A,n} \rangle_t & \langle c^\dagger_{A,m} c_{B,n} \rangle_t \\ \langle c^\dagger_{B,m} c_{A,n} \rangle_t & \langle c^\dagger_{B,m} c_{B,n} \rangle_t \end{pmatrix}. \tag{10}$$

In particular, Eq. (3) translates to an equation of motion for $G$.

Relying on translational invariance, we can focus on the Fourier transform

$$G_{m,n}(t) = \frac{1}{L} \sum_k e^{ik(m-n)} G_k(t), \qquad G_k(t) = \begin{pmatrix} \langle c^\dagger_{A,k} c_{A,k} \rangle_t & \langle c^\dagger_{A,k} c_{B,k} \rangle_t \\ \langle c^\dagger_{B,k} c_{A,k} \rangle_t & \langle c^\dagger_{B,k} c_{B,k} \rangle_t \end{pmatrix}. \tag{11}$$

Eq. (11) states that $G_k(t)$ is the generating symbol for $G_{m,n}(t)$, which is a block Toeplitz matrix. This fact will lead to finding a closed expression for the stationary state entanglement entropy using the Szegö theorem.

The time evolution Eq. (3) induces the equation of motion for $G_k^{X,Y}$ $(X, Y \in \{A, B\})$

$$\partial_t G_k^{X,Y} = i \langle H^\dagger_{\text{eff}} c^\dagger_{X,k} c_{Y,k} - c^\dagger_{X,k} c_{Y,k} H_{\text{eff}} \rangle_t - i \langle H_{\text{eff}} - H^\dagger_{\text{eff}} \rangle_t G_k^{X,Y}. \tag{12}$$

A straightforward computation lead to the exact correlation matrix [67]

$$G_k(t) = G_k(0) + \frac{1}{\mathcal{N}_k(t)} \begin{pmatrix} B_k(t) & e^{ik/2} A_k(t)(-C_k + 2JiD_k) \\ -e^{-ik/2} A_k(t)(C_k + 2JiD_k) & -B_k(t) \end{pmatrix}, \tag{13}$$

where we have defined

$$G_k(0) = \frac{1}{2} \begin{pmatrix} 1 & e^{ik/2} \\ e^{-ik/2} & 1 \end{pmatrix}, \quad \mathcal{N}_k(t) = 1 + (1 + C_k) A_k(t),$$

$$A_k(t) = \frac{\gamma^2 - h^2 \sin(k/2)^2}{2|E_k|^2} (1 - \cos(2E_k t)), \qquad B_k(t) = \frac{\gamma - h \sin(k/2)}{2|E_k|} \sin(2E_k t), \tag{14}$$

$$C_k = \frac{\gamma - h \sin(k/2)}{\gamma + h \sin(k/2)}, \quad D_k = \frac{\cos(k/2)}{\gamma + h \sin(k/2)}.$$

The dynamics of correlations in the non-Hermitian SSH was studied in Ref. [66, 67] revealing the emergence of supersonic modes propagating with integer multiples of the Fermi velocity. Here we focus on the dynamics of entanglement entropy that can be obtained directly from the correlation matrix. Given a bipartition $A \cup B$ the entanglement entropy is defined as $S = -\text{Tr}(\rho_A \log \rho_A)$ with $\rho_A = \text{Tr}_B(\rho)$. With a pure state $|\Psi(t)\rangle$, the entanglement entropy is computable from the spectrum of $G_{m,n}^A(t) = G_{m,n}(t)$ for $m, n \in A$. We consider $B$ an infinite system and $A$ a finite interval of length $\ell$. From the reduced correlation matrix, the entanglement is given by [75–77]

$$S_A = \sum_n s(\nu_n), \qquad s(x) \equiv -x \ln x - (1-x) \ln(1-x), \tag{15}$$

where $\nu_n$ are the eigenvalues of $G^A$. Eq. (15) is efficiently implementable and requires polynomial computational resources, whereas the Hilbert space would scale exponentially in system size. It requires computing the matrix elements of $G^A$ through Eqs. (11) and (13), and its diagonalization.

In Fig. 3, we present the time evolution of the entanglement entropy in the thermodynamic limit, for different subsystem sizes $\ell$ and three different values of $\gamma$, along the line $h = 1$. For

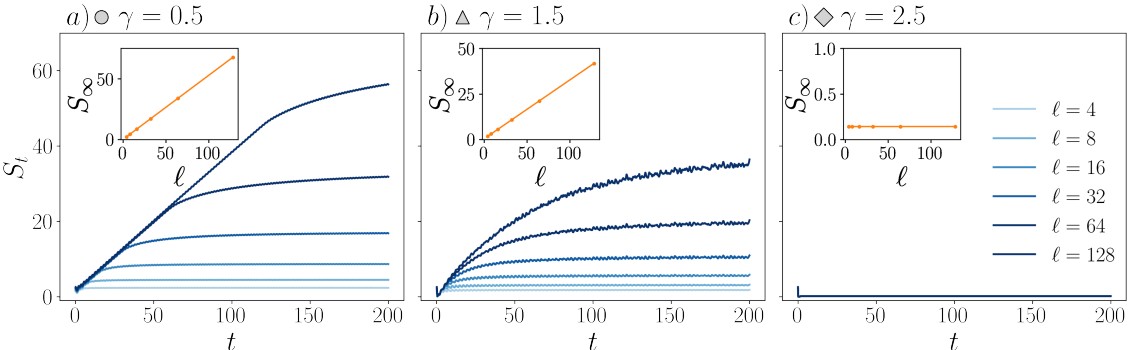

Figure 3: Time evolution of entanglement entropy in the thermodynamic limit, for different subsystem size $\ell$ and for three increasing values of $\gamma = 0.5, 1.5, 2.5$ along the $h = 1$ line (symbols in the top left corners corresponding to points of the phase diagram in Fig. 1). The dynamic is obtained by diagonalizing numerically the exact correlation matrix of the subsystem of size at each time step. (a-b) The growth is linear in time and saturates to a value $S_\infty$ which scale linearly with the subsystem size (inset), (c) The entanglement quickly settles to a small constant value $S_\infty$ which do no depend of system size (inset).

small values of the monitoring strength $\gamma$ (panel a, $\gamma = 0.5$) the entanglement grows linearly in time and saturates to a long-time value which scales with the size of the subsystem (see inset), corresponding to a volume-law scaling [67]. This behavior, similar to the unitary case [3], essentially persists upon increasing the value of $\gamma$ even beyond the $\mathcal{PT}$-symmetry breaking transition at $\gamma_{PT} = 1$. In fact, as we see in panel b for $\gamma = 1.5$, the entanglement entropy still grows in time, although slower than in the $\mathcal{PT}$-symmetric phase and possibly with a different dynamical behavior, and saturates to a steady-state value showing a clear volume-law scaling (see inset). A qualitative change in the behavior of the entanglement entropy occurs instead when $\gamma$ is increased further above $\gamma_c = 2$ (panel c). Here we see that the dynamics rapidly reaches a stationary state which is independent on subsystem size, compatibly with an area law scaling as expected on the Zeno side of an entanglement transition. Our numerical results on the entanglement dynamics shows therefore that a volume to area law entanglement transition emerges in the non-Hermitian SSH as a function of the measurement strength $\gamma$. In the next subsection we will confirm this result by computing analytically the leading entanglement entropy contribution in the large subsystem size limit $\ell \to \infty$.

## 4.2 Entanglement entropy of the stationary state

The results of previous section for the entanglement entropy were obtained numerically for a subsystem of size $\ell$ in a thermodynamically large chain. Here we show that the leading large-$\ell$ entanglement contribution can be obtained in closed form using manipulations similar to ones done for the unitary case [3].

To begin, we recast Eq. (15) for the entanglement entropy using the Cauchy theorem

$$S_A = \frac{1}{4\pi i} \oint_{\mathcal{C}} d\lambda\, e(0^+, \lambda) \frac{d}{d\lambda} \ln \det \tilde{G}^A(\lambda), \tag{16}$$

where, the contour $\mathcal{C}$ includes all the zeros of $\det \tilde{\Gamma}$, which all lie in the interval $[0, 1]$ and using

the $\ell$-dimensional identity matrix $\mathbb{1}_\ell$, we defined

$$\tilde{G}^A(\lambda) = \lambda\mathbb{1}_{2\ell} - G^A(\infty), \qquad G^A(\infty) \equiv \lim_{\tau\to\infty}\lim_{t\to\infty}\frac{1}{\tau}\int_t^{t+\tau}d\tau\, G^A(t),$$

$$\tilde{G}_k(\lambda) = \lambda\mathbb{1}_2 - G_k(\infty), \qquad G_k(\infty) \equiv \lim_{\tau\to\infty}\lim_{t\to\infty}\frac{1}{\tau}\int_t^{t+\tau}d\tau\, G_k(t), \qquad (17)$$

$$e(y,x) = -(y+x)\ln(y+x) - (1+y-x)\ln(1+y-x).$$

The key observation is that $\tilde{G}^A$ is a block Toeplitz matrix with generator $\tilde{G}_k$. The asymptotic form of $\mathcal{D}^A(\lambda) \equiv \ln\det\tilde{G}^A(\lambda)$ then is given by Szegö theorem

$$\mathcal{D}^A(\lambda) \simeq_{\ell\gg1} \ell\int_{-\pi}^{\pi}\frac{dk}{2\pi}\ln\det\tilde{G}_k. \qquad (18)$$

After simple manipulations, we have

$$\frac{d}{d\lambda}\mathcal{D}^A(\lambda) = \ell\int_{-\pi}^{\pi}\frac{dk}{2\pi}\frac{2\lambda-1}{(\lambda-\nu_+(k))(\lambda-\nu_-(k))}, \qquad (19)$$

with $\nu_\pm(k)$ the eigenvalues of $G_k$. In the stationary limit these eigenvalues can be exactly computed and depend on the energy $E_k$ being real or imaginary. When $E_k$ is real

$$\nu_\pm(k) = \frac{1\pm v_k}{2}, \qquad v_k = \sqrt{4\chi_k^2\left(C_k^2+4D_k^2\right)-4\chi_k C_k+1}, \qquad (20)$$

where we have introduced the auxiliary functions

$$A_k^\infty \equiv \frac{\gamma^2-h^2\sin^2(k/2)}{2|E_k|^2},$$

$$\chi_k \equiv \lim_{\tau\to\infty}\lim_{t\to\infty}\frac{1}{\tau}\int_t^{t+\tau}d\tau\frac{A_k(t)}{N_k(t)} = \frac{1}{1+C_k}\left(1-\frac{1}{\sqrt{2(1+C_k)A_k^\infty+1}}\right). \qquad (21)$$

Since $\nu_{k,\pm}\neq0$, these eigenvalues contribute in Eq. (19) and hence in Eq. (16). Conversely, for imaginary $E_k$ we have $\nu_+=1$ and $\nu_-=0$, and as a result imaginary modes do not contribute to the stationary state volume-law entanglement.

Evaluating the integral we obtain a closed form expression for the leading behavior of the entanglement entropy which reads The final expression for the entanglement entropy reads therefore $S_A = v(h,\gamma)\ell + O(1)$ where the slope $v(h,\gamma)$ of the volume-law entanglement contribution reads

$$v(h,\gamma) = \int_{-\pi}^{\pi}\frac{dk}{2\pi}\Theta\left(h^2-\gamma^2+(4-h^2)\cos^2\left(\frac{k}{2}\right)\right)s(\nu_{k,+}). \qquad (22)$$

From this result we see that $v(h,\gamma)$ depends on system parameters and it is written in closed form in term of an integral of an entropy function $s(\nu_{k,+})$, with the integration domain restricted to those momenta with purely real-eigenvalues, i.e. those enclosed within the pair of EPs in the spectrum (see Fig. 2).

We plot this quantity and compare it with the numerical late time one in Fig. 4 (left panel) for $h=1$, finding perfect agreement. This result confirms the entanglement transition from volume to area law, already evoked from the dynamics, which is sharply characterized here by the vanishing of the slope coefficient $v(h,\gamma)$ at $\gamma_c=2$ for $h=1$. From our exact result we can see that the vanishing of the slope coefficient $v(h,\gamma)$ is in part driven by the merging of the

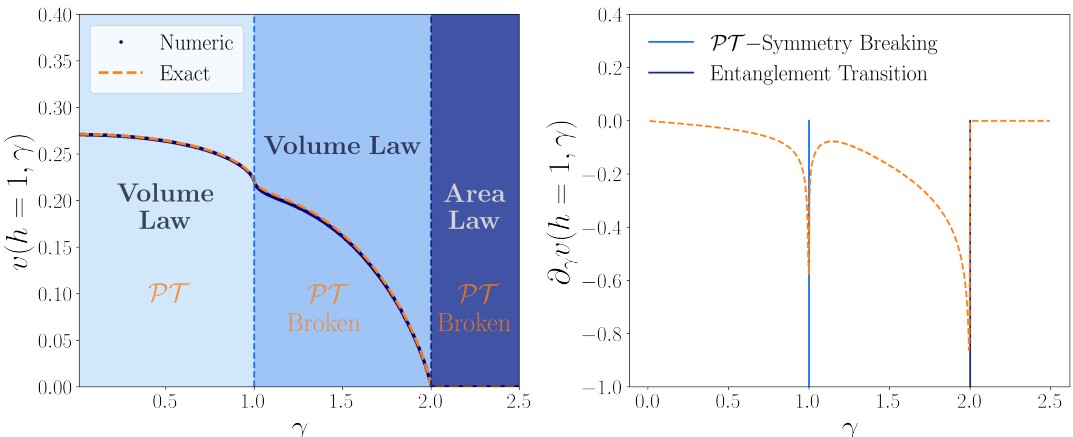

Figure 4: Left-Panel: prefactor of the volume law entanglement entropy $v(h, \gamma) = S_A/\ell$ as a function of $\gamma$ along the line $h = 1$ as obtained from the numerical evaluation and the analytical formula in Eq. (22). We notice a perfect agreement between the two methods, and a clear volume-to-area law transition at $\gamma_c = 2$. Right-Panel: Derivative $\partial_\gamma v(h = 1, \gamma)$ presenting interestingly two divergences, first at $\gamma_{PT} = 1$ when the $\mathcal{PT}$-symmetry breaking occurs and then at $\gamma_c = 2$ when the volume-to-area law transition happens.

exceptional points $k_{EP}(h, \gamma)$ at the gapless to gapped transition $\gamma_c$, when the support of the integral in Eq. (22) shrinks to zero as $k_{EP}(h, \gamma) \sim \sqrt{\gamma_c - \gamma}$, and in part by the vanishing of the entropy function as $v_{k,+} \to 1$, resulting in a linear behavior near $\gamma_c$ as seen in Fig. 4.

This volume-to-area entanglement transition is clearly separated from the $\mathcal{PT}$-symmetry breaking point, which occurs for $h = 1$ at $\gamma_{PT} = h = 1$. Interestingly, we see that nevertheless a non-analytic behavior emerges at $\gamma_{PT}$, which can be interpreted as a weaker volume-to-volume "transition". This is identified by looking at the derivative $\partial_\gamma v(h, \gamma)$ which diverges at $\gamma_{PT}$ as we show in the right panel of Fig. 4.

## 5 Discussion

In this section we discuss our results, comment on their broader implications for the entanglement properties of non-unitary quantum many-body systems and on related results in the literature.

Our main result is the existence of a volume-to-area law entanglement transition in the non-Hermitian SSH model and its analytical characterization in terms of the spectral properties of the system. As our exact results show clearly, the origin of this transition can be qualitatively understood in terms of quasiparticles acquiring a finite life-time due to the measurement back-action and thus not contributing to the volume law scaling (See Eq. 22). While this process starts at the $\mathcal{PT}$-symmetry breaking, when first imaginary parts appear in the spectrum, it is only when all the quasiparticle modes become short-lived and a gap opens up in the spectrum of decay modes (imaginary part of the complex energy) that a true entanglement transition to an area law arises. One could therefore wonder whether a similar mechanism could hold more generally beyond the SSH case, for non-Hermitian quantum many-body systems with $\mathcal{PT}$-symmetry breaking. In this respect we could speculate that as long as the $\mathcal{PT}$-symmetry breaking occurs *gradually*, with imaginary parts remaining zero for a finite density of modes, then a volume law phase could be sustained since these mode would dephase and heat up. On the other hand, if the spectrum at the $\mathcal{PT}$-symmetry breaking acquires a finite imaginary

part leading to a gap in the spectrum of decay modes, then we could expect the entanglement transition into an area law to coincide with the $\mathcal{PT}$-symmetry breaking.

Our results for the dynamics of non-Hermitian SSH model complement the results of Ref. [67], which had focused only on a quench at the $\mathcal{PT}$-symmetry breaking point finding volume law scaling. Furthermore it is worth emphasizing that that our results concern entanglement entropy *dynamics* after a quench of the dissipation and thus are different than those obtained in Ref. [72]. This work in fact takes a different approach to non-Hermitian systems based on bi-orthogonal quantum mechanics and therefore focuses on the entanglement entropy of low-energy eigenstates of the model in the $\mathcal{PT}$-symmetric phase or at the critical points. This results in a logarithmic scaling of the entanglement and negative central charges as opposed to our volume vs area law scaling.

Finally, it is worth mentioning that the non-Hermitian version of the SSH model discussed here, including the $\mathcal{PT}$-symmetry breaking, has been experimentally implemented in the field of topological photonics, in particular using photonic waveguide arrays [78–80]. There the non-Hermitian quantum dynamics is simulated as propagation along the axial direction of the waveguide, using the analogy between paraxial Maxwell equations and Schrödinger equation. A staggered imaginary chemical potential in the two-sublattices can be easily implemented using alternating gain and losses. It could be interesting to discuss whether an analog dynamics for the correlation matrix could be achieved with these platforms, which could give access to the dynamics of the entanglement entropy.

# 6 Conclusion

In this work we have studied the entanglement dynamics in a non-Hermitian SSH free fermionic model, arising as the no-click limit of a quantum jump master equation. We analytically find two types of critical behavior: a spectral $\mathcal{PT}$ symmetry breaking transition and a volume-to-area law entanglement transition for the stationary state. Importantly, the two do not coincide, despite at the $\mathcal{PT}$-phase transition the volume law entanglement prefactor exhibits singular behavior.

Several open questions remain to be addressed and will be the subject of future investigations. For what concerns the non-Hermitian SSH the exact time evolution of the entanglement entropy can possibly be computed in the scaling limit, following the arguments in Ref. [81]. Furthermore it would be interesting to discuss the entanglement dynamics of this model for open-boundary conditions. From one side it is known that the $\mathcal{PT}$-symmetric phase is a topological non-Hermitian phase [63, 72] hosting boundary modes and characterized by a non-zero complex Berry phase, therefore it would be interesting to see whether signatures of this non-trivial topology emerge in the dynamics of the entanglement. Similar questions have been raised for monitored quantum systems [46, 47]. In addition, it is known that for non-Hermitian systems changing the boundary conditions can have important effects on the physics of the problem, including entanglement properties [70].

The existence of a volume to area law entanglement transition in a free non-Hermitian model is an interesting result by itself, which suggests that the patterns of entanglement in these systems are richer than in the unitary case and calls for further studies. The results of this work, together with those obtained for the non-Hermitian Ising chain [43], point towards a close connection between spectral properties of the system and entanglement dynamics which would be interesting to put on a firmer grounds through a phenomenological quasiparticle picture for entanglement in non-Hermitian systems. We note that a similar phenomenology of entanglement transitions has been recently obtained in non-unitary gaussian circuit with spatial and time periodicity [82].

Finally, the scaling of the entanglement in the complete quantum jumps protocol beyond the no-click limit needs to be studied. In particular we would like to understand whether an entanglement transition will survive in that case and if the error-correcting phase will preserve a volume-law entanglement entropy. This would be particularly interesting in light of the recently proposed quasiparticle picture for monitored quantum many-body systems [43].

## Acknowledgments

**Funding information** We acknowledge support from the ANR grant "NonEQuMat" (ANR-19-CE47-0001) and computational resources on the Collège de France IPH cluster.

## A  Non-Hermitian Quantum Quenches from Quantum Jumps

In this appendix we discuss how the non-Hermitian SSH model that we considered in the main text arises as the no-click limit of a monitored SSH model. Specifically we consider a conventional SSH chain with Hamiltonian $H$ defined in the main text, coupled to local measurement apparatus. These continuously monitor, stochastically and independently, the local density of particles on sublattice $A$, $n_{A,i} = c_{A,i}^\dagger c_{A,i}$, and the local density of holes, $1 - n_{B,i} = c_{B,i} c_{B,i}^\dagger$, on sublattice $B$. We consider a quantum jump protocol, where the evolution of the system is described by the stochastic Schrödinger equation (see e.g. [71] and references therein for a detailed derivation)

$$
\begin{aligned}
d|\Psi(t)\rangle = &-iH dt|\Psi(t)\rangle - i\frac{dt}{2}\langle H_{\text{eff}} - H_{\text{eff}}^\dagger \rangle_t |\Psi(t)\rangle \\
&+ \sum_{i=1}^{L}\left[ dN_{A,i}^t \left( \frac{n_{i,A}}{\sqrt{\langle n_{i,A}\rangle_t}} - 1 \right) + dN_{B,i}^t \left( \frac{1 - n_{i,B}}{\sqrt{\langle 1 - n_{i,B}\rangle_t}} - 1 \right) \right]|\Psi(t)\rangle,
\end{aligned}
\tag{A.1}
$$

where $\langle \circ \rangle_t \equiv \langle\Psi(t)| \circ |\Psi(t)\rangle$ is the expectation value, $dN_{A,i}^t, dN_{B,i}^t \in \{0,1\}$ are independent Poisson processes with $\overline{dN_{X,i}} = \delta p_{X,i}$, $dN_{X,i}dN_{Y,i} = \delta_{i,j}\delta_{X,Y}dN_{X,i}$ with $X,Y \in \{A,B\}$. The probability of a quantum jump are uncorrelated for any site and fermionic type and given by

$$
\delta p_{A,i} = 2\gamma \delta t \langle\Psi(t)|c_{A,i}^\dagger c_{A,i}|\Psi(t)\rangle, \quad \text{and} \quad \delta p_{B,i} = 2\gamma \delta t \langle\Psi(t)|c_{B,i}c_{B,i}^\dagger|\Psi(t)\rangle.
\tag{A.2}
$$

In Eq. (A.1) we have introduced the non-Hermitian Hamiltonian

$$
H_{\text{eff}} = H - ih \sum_{i=1}^{L}\left( c_{A,i}^\dagger c_{A,i} - c_{B,i}^\dagger c_{B,i} \right).
\tag{A.3}
$$

This is the evolution that follows the post-selected trajectory $dN_{X,i}^t = 0$ for all $t, i$ and $X = A, B$. In this case, the evolution is deterministic and given by

$$
|\Psi(t)\rangle = \frac{e^{-iH_{\text{eff}}t}|\Psi(0)\rangle}{\|e^{-iH_{\text{eff}}t}|\Psi(0)\rangle\|}.
\tag{A.4}
$$

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
