# Peer review of "Volume-to-Area Law Entanglement Transition in a non-Hermitian Free Fermionic Chain"

_SciPost Physics, doi:SciPost Phys. 14, 138 (2023)_

## Round 1 · Referee Report · Anonymous (Referee 1) · 2022-12-3

"Volume-to-Area Law Entanglement Transition in a non-Hermitian Free Fermionic Chain" by Youenn Le Gal, Xhek Turkeshi, Marco Schirò

While many studies on entanglement entropies in non-Hermitian systems have been carried out, and in particular for the Su-Schrieffer-Heeger model, the emphasis of this paper is original. The authors compare the transition from a volume to an area scaling law in comparison to the transition from the PT-symmetric to the spontaneously broken PT-symmetry regime. The overall conclusion is that the transitions are not directly related as the volume to an area scaling law takes place deep inside the broken PT-symmetry regime. In fact the transition occurs precisely at the onset of the gapless spectrum. The authors speculate that only when all the short lived quasi-particles are present the transition occurs. This is an interesting results that inspires further investigations to clarify the universality of this observation.

The presentation is in general clear, but I have a few, mostly minor, points the authors should address:

- I think it is important to properly define the norm used in equation (3). In the PT-symmetric regime one would expect the norm to be conserved in contrast to the general statement made after this equation. Is this not the case?

- Equation (4) is not what is usually referred to as the nonlinear Schrödinger equation. The authors should also say in which sense this equation is meant to be nonlinear.

- "Schrödinger equation" appears in all kinds of variants throughout the text, mostly as Schrodinger equation, but also as Schrïdinger equation. This should be fixed.

- The labelling between letters in figures and the text should be made consistent, e.g. in figure 1 the authors use $\phi$ whereas in equation (8) the letter $\Phi$ is used.

- In figures 2 and 3 some circles, triangles and diamond appear that have no use.

- There should be a comment on how the contour in equation (16) is to be understood.

- The references need some tidying up, for instance [55] and [73] are identical.

- The following reference seems to be relevant: Ali, T., Bhattacharyya, A., Haque, S. S., Kim, E. H., Moynihan, N. (2020). Post-quench evolution of complexity and entanglement in a topological system. Physics Letters B, 811, 135919.

After these issues have been addressed properly I would like to recommend the manuscript for publication.

---

## Round 1 · Referee Report · Anonymous (Referee 2) · 2022-12-20

Strengths

1 . The paper demonstrates that a volume-to-area law transition in the entanglement entropy exists in a certain free fermionic system subject to a continuous monitoring via exact computations

  1. It also connects the transition with a PT symmetry breaking that accompanies gap-opening.

  2. Analytic results are verified by numerics.

Weaknesses

Discussions are limited to the free model (SSH model).

Report

The present article studies the entanglement dynamics in a non-Hermitian free fermionic system, which arises as the particular limit of a monitored version of it. The main finding in this paper is that the volume-to-area law transition of the entanglement entropy is triggered by a PT-symmetry breaking that leads to a gap. Interestingly, such transition is prevented for another spectral transition that simply generates gapless modes with the imaginary part. As far as I know, these different behaviours of the entanglement transition depending on the nature of PT-symmetry breaking hasn't known, and deserves further investigations. I recommend publication for this article, but I have a few suggestions which authors can choose to address.

Requested changes

  1. It is not entirely clear to me why it is interesting to deal with the SSH model. It would be nice to have a brief comment to motivate the choice of this model. In particular, is there any aspect of this model that makes it somewhat different from other free fermionic systems, e.g. the Kitaev chain?

  2. In the paper it is explained as to what kind of PT-symmetry breaking points induce a volume-to-area law transition at the level of spectrum, but it would be interesting to understand what the difference of these points mean in terms of quantum jumps.

  3. There is a typo in the second paragraph of p.11 ("Furthermore tt"... should be replaced by "Furthermore it"...)

---

## Round 2 · Author Response

The main changes in this new version are : - Improvement of the discussion on the non-Hermitian SSH model and on the motivations for our study. - Clarification of the role of the normalization in the non-Hermitian evolution.

---

## Round 2 · List of Changes

We thank Referee A for their report on our work and the overall positive assessment. - " I think it is important to properly define the norm used in equation (3). In the PT-symmetric regime one would expect the norm to be conserved in contrast to the general statement made after this equation. Is this not the case?" The normalization of the wave-function under non-Hermitian evolution is directly inherited from the no-click limit of the quantum jump dynamics (discussed in Appendix A). We note that also in the P T −symmetric phase, where all eigenvalues are real, the norm is not necessarily conserved because in general eigenstates of non-Hermitian Hamiltonian are not orthogonal. We have added a clarification in the text on this point. - "Equation (4) is not what is usually referred to as the nonlinear Schrodinger equation. The authors should also say in which sense this equation is meant to be nonlinear." We agree that the use of non-linear could be confusing and we have modified in the text accordingly. To clarify, here we meant that the non-Hermitian evolution is not linear because it depends on the state itself (through the norm), but this should not be confused with non-linearities arising from interactions. - "Schrodinger equation appears in all kinds of variants throughout the text, mostly as Schrodinger equation, but also as Schrodinger equation. This should be fixed" We have fixed the notation. • "The labelling between letters in figures and the text should be made consistent, e.g. in figure 1 the authors use φ whereas in equation (8) the letter Φ is used." We fixed the notation. •" In figures 2 and 3 some circles, triangles and diamond appear that have no use" Symbols in figures 2 and 3 correspond to the points in the phase diagram of figure 1, to make clear where are we in the phase diagram of the system. We have clarified this point in the caption of the figures 2-3. •" There should be a comment on how the contour in equation (16) is to be understood" We have added a comment on this point. •" The following reference seems to be relevant: Ali, T., Bhattacharyya, A., Haque, S. S., Kim, E. H., Moynihan, N. (2020). Post-quench evolu- tion of complexity and entanglement in a topological system. Physics Letters B, 811, 135919." We have added this reference
We thank Referee B for their comments which we seriously took into account to improve the manuscript. -" It is not entirely clear to me why it is interesting to deal with the SSH model. It would be nice to have a brief comment to motivate the choice of this model. In particular, is there any aspect of this model that makes it somewhat different from other free fermionic systems, e.g. the Kitaev chain?" We thank the Referee for this comment, we have expanded the introduction and the discussion of this model to clarify our motivations. The main interest in studying this specific SSH model is that it satisfies PT symmetry and displays PT symmetry breaking, which allows us to study entanglement transitions and to connect them to spectral transitions in the model. In this respect the topological aspects of the problem are not crucial here. We have also added some references on the non-Hermitian Kitaev chain. If a PT symmetric version of this model would be considered we could expect similar results. - "In the paper it is explained as to what kind of PT-symmetry breaking points induce a volume-to-area law transition at the level of spectrum, but it would be interesting to understand what the difference of these points mean in terms of quantum jumps." This is a very interesting and open question. We could speculate that in presence of quantum jumps an entanglement transition would persist but possibly the volume law phase could turn into a sub-volume (logarithmic) critical phase. Answering to this question is however beyond the scope of this work and is left for future studies. "- There is a typo in the second paragraph of p.11 (”Furthermore tt”... should be replaced by ”Furthermore it”...)" We have fixed this typo and other misprints throughout the manuscript.

Anonymous on 2023-02-19 [id 3376]
The authors have addressed all points I have raised, but there is still an issue with the answer to my first question regarding the norm in equation (3). One usually strictly distinguishes between open and PT-symmetric systems. The whole point of PT-symmetric quantum mechanics is that in the PT-symmetric regime one can define a new inner product that allows for a unitary evolution, which is not possible to construct in the broken regime. The authors should clearly state that they do not follow the approach advocated in reference [55] and further developed in many papers thereafter. It would also be useful to readers familiar with that more than two decades old field of research to provide a reasoning on why they chose to ignore the insight gained in that field and justify their approach.
Anonymous on 2023-02-23 [id 3395]
(in reply to Anonymous Comment on 2023-02-19 [id 3376])To clarify our approach in this paper, we study the non-unitary dynamics (non-Hermitian Hamiltonian evolution), which is the deterministic part of the quantum jumps protocol (no-click limit). Thus, we do not aim to revert back to unitary evolution, even if PT-symmetry allows it. In particular changing the inner product would not give us a new insight on the dynamics since it would imply doing the same procedure as before on the broken symmetry part with this new inner product. Our interest is to show how PT-symmetry breaking modifies the non-unitary dynamics but the dynamical behavior we consider deals only with right eigenvectors, and thus we do not use any bi-orthogonal quantum mechanics properties that PT symmetry can allow.
We apologize for any confusion that may have been caused by not clearly stating this in the paper. We will modify the manuscript to clarify this point. Thank you again for your feedback.

---

## Round 3 · List of Changes

Footnote added to clarify the framework in which the work is done.

---

## Editorial Decision

published